# Identification and Characterization of Three Heat Shock Protein 90 (Hsp90) Homologs in the Brown Planthopper

**DOI:** 10.3390/genes11091074

**Published:** 2020-09-12

**Authors:** Xuan Chen, Ze-Dong Li, Yi-Ting Dai, Ming-Xing Jiang, Chuan-Xi Zhang

**Affiliations:** 1State Key Laboratory of Rice Biology and Ministry of Agriculture Key Laboratory of Agricultural Entomology, Institute of Insect Science, Zhejiang University, Hangzhou 310058, China; xuanchen@zju.edu.cn (X.C.); lizedong@zju.edu.cn (Z.-D.L.); 21716092@zju.edu.cn (Y.-T.D.); 2State Key Laboratory for Managing Biotic and Chemical Threats to the Quality and Safety of Agro-Products, Key Laboratory of Biotechnology in Plant Protection of MOA of China and Zhejiang Province, Institute of Plant Virology, Ningbo University, Ningbo 315211, China

**Keywords:** Hsp90, RNA interference, development, oogenesis, cuticle structure, *Nilaparvata lugens*

## Abstract

Hsp90 (heat shock protein 90) chaperone machinery is considered to be a key regulator of proteostasis under both physiological and stress growth conditions in eukaryotic cells. The high conservation of both the sequence and function of Hsp90 allows for the utilization of various species to explore new phenotypes and mechanisms. In this study, three Hsp90 homologs were identified in the brown planthopper (BPH), *Nilaparvata lugens*: cytosolic *NlHsp90*, endoplasmic reticulum (ER) *NlGRP94* and mitochondrial *NlTRAP1*. Sequence analysis and phylogenetic construction showed that these proteins belonged to distinct classes consistent with the predicted localization and suggested an evolutionary relationship between *NlTRAP1* and bacterial HtpG (high-temperature protein G). Temporospatial expression analyses showed that *NlHsp90* was inducible under heat stress throughout the developmental stage, while *NlGRP94* was only induced at the egg stage. All three genes had a significantly high transcript level in the ovary. The RNA interference-mediated knockdown of *NlHsp90* its essential role in nymph development and oogenesis under physiological conditions. *NlGRP94* was also required during the early developmental stage and played a crucial role in oogenesis, fecundity and late embryogenesis. Notably, we first found that *NlHsp90* and *NlGRP94* were likely involved in the cuticle structure of female BPH. Together, our research revealed multifunctional roles of Hsp90s in the BPH.

## 1. Introduction

Hsp90 (heat shock protein 90), with an approximate molecular weight of 90 kDa, is a highly ubiquitous molecular chaperone, which accounts for 1–2% of cellular proteins under normal growth conditions [1]. Hsp90 protein family members stabilize, activate and regulate target proteins together with a cohort of cochaperones under both physiological and stress conditions, maintaining proteostasis and thus cellular homeostasis [2,3].

Apart from molecular weight, further classification of Hsp90 can be based on subcellular localization [4]. Most bacteria have one homolog of Hsp90, known as HtpG (high-temperature protein G). However, HtpG is largely absent from Archaea [5]. Higher eukaryotes harbor up to four distinct homologs of Hsp90: Hsp90 in the cytosol, GRP94 (94 kDa glucose-regulated protein) in the endoplasmic reticulum (ER), mitochondrial TRAP1 (tumor necrosis factor receptor-associated protein 1), and chloroplast Hsp90C (only exists in plants) [6].

Hsp90 proteins are highly conserved in all eukaryotes as well as in bacteria. Typically, an HSP90 monomer has three highly conserved domains: the N-terminal domain (NTD), middle domain (MD) and C-terminal domain (CTD) [7]. Besides, the C-terminal motif Met-Glu-Glu-Val-Asp (MEEVD) is an typical characteristic of the cytosolic Hsp90, which is of vital importance in the interaction with cochaperones containing tetratricopeptide repeat (TPR) domains [8]. In addition, an extended, flexible, charged linker connects the NTD and the MD, which not only modulates NTD–MD contacts but also affects Hsp90 function [8,9].

Previous studies have suggested that, in insects, most Hsp90s are cytosolic or ER based [5,6,10]. Mammals and yeast generally have two closely related Hsp90 isoforms present in the cytosol, while most invertebrates, such as *Drosophila melanogaster* and *Caenorhabditis elegans,* contain only one cytosolic Hsp90 [1,11]. The *Drosophila* Hsp90 has been demonstrated to be in the involvement of the development of muscle, germ cells and multiple phenotypic traits, such as wing, antenna and eye defects [6]. Similarly, studies in *Tribolium castaneum* have shown that Hsp83 (a cytosolic member) contributes to oogenesis, compound eye development, and heat resistance [12,13].

The ER isoform of Hsp90, usually called GRP94 (94-kDa glucose-regulated protein), is the most abundant glycoprotein in the ER and is therefore also known as endoplasmin [14]. As an ER-resident chaperone, GRP94 contains an N-terminal ER retention signal required for ER targeting [2,6]. In contrast to cytosolic Hsp90, there is generally a KDEL retention signal rather than a MEEVD motif in the C-terminus of GRP94 [7]. The major function of GRP94 is to facilitate the folding as well as the assembly of secretory and membrane proteins [15].

ER-based Hsp90 is required during the early developmental stage in *C. elegans*, *Drosophila*, and mice [15,16,17]. Gp93 (GRP94 ortholog) is an essential gene in *Drosophila*. The loss of Gp93 expression led to lethal phenotype in late larvae, and mutant larvae defected severely in the gut epithelium along with a starvation-like phenotype [16]. In addition, ER-based Hsp90 is important for larval/pupal development, lifespan, and reproduction in *T. castaneum* in addition to heat stress response [18]. Apart from the above-mentioned discoveries; however, there are still limited studies that evaluated the function of the GRP94 ortholog in insects.

Identical to heat shock protein 75 (Hsp75), TRAP1 belongs to the Hsp90 family. It has been indicated to bind to the retinoblastoma protein during mitosis and after heat shock [19]. As the mitochondrial member of Hsp90, TRAP1 contains an N-terminal mitochondria-targeting sequence but lacks both the C-terminal MEEVD motif and the charger linker [20,21]. TRAP1 has been shown to protect cells from oxidative stress and apoptosis [22]. In *Drosophila*, knockdown of TRAP1 led to mitochondrial dysfunction and an increase in sensitivity to stress [23]. However, few studies have identified TRAP1 or illustrated its function in insects other than *Drosophila*.

Most recently, a cytosolic Hsp90 was identified in the brown planthopper (BPH), *Nilaparvata lugens*, one of the most destructive pests of rice in Asia, and the study showed that *Nl*Hsp90 is involved in resistance to temperature stress in two wing morphs of *N. lugens* [24]. In this study, we further investigated the more detailed and novel biological functions of the cytosolic Hsp90 in BPH. Moreover, two new Hsp90 homologs that are predicted to be located in the ER and mitochondria were also identified. The gene structure, phylogenetic relationship, expression patterns and RNA interference (RNAi) effects were analyzed to further understand the potential biological roles of these Hsp90 homologs in BPH. Our results illustrate that Hsp90s play multifunctional roles in the development, oogenesis, fecundity, late embryogenesis and cuticle structure in *N. lugens*.

## 2. Materials and Methods

### 2.1. Insects

The BPH population used in this study were originally obtained from Hangzhou, China. The insects were reared on rice seedlings (rice strain: Xiushui 134) in a walk-in chamber at 27 ± 0.5 °C with 60 ± 5% relative humidity. The photoperiod was set as 16 h:8 h (light:dark).

### 2.2. Identification of Hsp90 Homologs in BPH

The Hsp90 homologs in BPH were identified using bioinformatics approaches. Briefly, the amino acid sequences of the Hsp90 family genes in *D. melanogaster* obtained from FlyBase (http://flybase.org/) were used as queries against the local BPH genomic and transcriptomic databases to find homologs in BPH. The open reading frame (ORF) was predicted on the Softberry website. Specific primers were designed based on the predicted ORFs to amplify the cDNA sequences of Hsp90 homologs, and all of the ORFs were confirmed by PCR. The primers used are shown in Appendix A.

Website InterPro (https://www.ebi.ac.uk/interpro/) and Cell-PLoc-2 (http://www.csbio.sjtu.edu.cn/bioinf/Cell-PLoc-2/) were used to predict putative domains and subcellular localization from three protein sequences, respectively. A phylogenetic tree was constructed by the MEGA X program with the neighbor-joining method. Bootstrap was set as 1000 replications to identify homologous relationships. The sequences that have been identified in 18 species were used in this analysis (GenBank accession numbers are provided in Appendix A).

### 2.3. Real-Time Quantitative PCR (RT-qPCR) Analysis

To examine whether *Nl*Hsp90s are inducible by thermal and cold stress, insects at different developmental stages (eggs, second instar nymphs, fifth instar nymphs and adults 24 h after emergence) were collected and reared at 4 °C, 27 °C and 37 °C for 1 h. Thereafter, the expression level was measured. Different tissue samples were dissected and collected from the integuments (50), fat bodies (50) and guts (50) of 5th instar nymphs. Testes (50) and ovaries (50) were collected from adult male and female BPHs, respectively.

After total RNA was extracted using RNAiso Plus (Takara, Kyoto, Japan) and treated with DNase I, 1 μg RNA was used to reverse-transcribe and synthesize first-strand cDNA using the HiScript^®^ II Q RT SuperMix (Vazyme, Nanjing, China). The target genes were quantified by real-time qPCR (RT-qPCR) with an SYBR Color qPCR Master Mix kit (Vazyme, Nanjing, China). The reaction was conducted on a CFX96TM real-time PCR detection system (Bio-Rad, Hercules, CA, USA) with thermal cycling program set as follows: denaturation for 3 min at 95 °C, followed by 40 cycles at 95 °C for 10 s and 60 °C for 30 s. Specific primers for RT-qPCR were designed using Primer Premier 6.0 software (Appendix A), and the housekeeping gene 18S rRNA was used as an internal control. The 2^-ΔΔCt^ method was used to evaluate the quantitative variation between the relative mRNA expression level of target genes and the reference gene. Each sample was analyzed with three independent biological replicates with three respective technical replications.

### 2.4. RNA Interference

Double-stranded RNAs (dsRNAs) of target genes were synthesized from the amplified sequences using a T7 high-yield transcription kit (Vazyme, Nanjing, China) The specific primers used for dsRNAs synthesis were listed in Appendix A. GFP was used as the negative control.

RNAi was carried out as described previously [25,26]. Briefly, 2nd- and 4th instar nymphs or newly emerged females were anaesthetized with carbon dioxide for 5–10 s. Then, approximately 25 ng, 50 ng and 100 ng of dsRNA was microinjected into the mesothorax of 2nd instar nymphs, 4th instar nymphs and females under a FemtoJet (Eppendorf-Netheler-Hinz, Hamburg, Germany), respectively. After injection, the insects were reared on fresh rice seedlings to recover. RNAi efficiency was evaluated three days after injection by RT-qPCR (6 to 8 insects were collected as one independent sample). The remaining insects were maintained for development observations and survival statistics.

For fecundity analysis, females were mated with virgin males three days after injection in a glass tube with fresh rice seedlings for 24 h. Then, the males were removed, and the females were allowed to oviposit for 5 days. The rice seedlings were maintained for another ten days to count the number of hatched offspring. Afterwards, we dissected the leaf sheaths of the rice seedlings under the microscope and recorded the number of eggs that failed to hatch. The total number of hatched offspring and the number of remaining eggs in the leaf sheaths were considered to be the oviposition amount. Each treatment included 10 biological replicates.

### 2.5. Transmission Electron Microscopy (TEM) Observation

To examine the ultrastructure, the dorsal abdominal integument of females six days after dsRNA treatment was dissected carefully under a microscope for observation. In detail, the newly emerged females were injected with dsRNA targeting different genes and reared under normal growth conditions. The dorsal abdominal integument between the third and fifth segments was dissected six days after injection. Preparation of samples was performed according to a previous report [27], and the treated specimen sections were observed under a Hitachi Model H-7650 Transmission Electron Microscope (TEM).

### 2.6. Statistical Analysis

The data are presented as the mean ± SEM (standard error of mean). Statistical analysis was performed using Microsoft EXCEL and GraphPad Prism 8.3.0. A two-tailed Student’s *t*-test was used to compare the difference between two groups. The significance level was set at * *p* < 0.05 and *** *p* < 0.001.

## 3. Results

### 3.1. Identification and Analysis of the Hsp90 Homolog Gene Structure in BPH

By using the known *D. melanogaster* Hsp83, Gp93 and TRAP1 proteins as query sequences to search the *N. lugens* genome and transcriptomic databases, three candidate Hsp90 homologs were finally identified. Based on the predicted molecular weight and subcellular location, these three Hsp90 homologs were named *Nl*Hsp90, *Nl*GRP94, and *Nl*TRAP1.

All three proteins have three typical conserved domain architectures (Figure 1), which indicates that they belong to the Hsp90 family. In detail, the *NlHsp90* gene contains a 2193 bp ORF interrupted by three introns. It encodes a protein of 730 amino acids (aa) in length, with a predicted molecular weight of 83.8 kD and an isoelectric point of 4.95. NlHsp90 has the typical characteristic structure of cytosolic *Hsp90* with three conserved domains containing the Met-Glu-Glu-Val-Asp (MEEVD) motif in the C-terminal domain (CTD). Besides, as shown in Figure 1, a charged linker connects the NTD and MD. The predicted gene structure was consistent with that of cytosolic Hsp90 proteins.

The second gene, *NlGRP94*, with a length of 2388 bp, includes twelve exons and eleven introns. The 795 aa protein encoded by this gene has a weight of 90.8 kDa and a pI of 4.88. In addition to the three conserved domains of Hsp90, *NlGRP94* contains an additional 23 aa signal peptide in the N-terminus and an HDEL in the C-terminus rather than the reported KDEL motif, which was thought to be the signature of endoplasmic reticulum localization [2].

Finally, the *NlTRAP1* gene is 1869 bp in length with eight exons and encodes a protein of 622 amino acids. The putative isoelectric point is 6.04 and the predicted molecular weight is 70.2 kDa. Conserved domain prediction shows that it contains an N-terminal mitochondria-targeting sequence but lacks both the C-terminal MEEVD motif and the charger linker.

### 3.2. Phylogenetic Analysis

To investigate the evolutionary relationships of different Hsp90 homologs, a phylogenetic analysis based on Hsp90 homologs from 18 species was performed and suggested that the three Hsp90s in BPH were classified into three distinct classes (Figure 2). We clearly observed that the cytosolic, ER and mitochondrial homologs of Hsp90 across diverse species were clustered separately. In the cytosolic Hsp90 class A, *Nl*Hsp90 shared high similarity with other known Hemiptera Hsp90s, and the dendrogram was consistent with the evolutionary relationship of taxonomic orders. However, *Nl*GRP94 was clustered alone into a branch, although close to those from other insects, such as *Bemisia tabaci* and *Halyomorpha haly*s, in Hemiptera, indicating that *Nl*GRP94 might have different functions. Interestingly, the mitochondrial homologs *Nl*TRAP1 were clustered with other insect TRAP1s and HtpG from bacteria in class B.

### 3.3. Temporospatial Expression Patterns

To better understand the potential role of Hsp90 homologs in BPH, the developmental expression pattern under temperature stress was first investigated using RT-qPCR. Under normal growth conditions, that is, at 27 °C, *NlHsp90* was widely expressed in eggs, early-instar nymphs (2nd), late-instar nymphs (5th) and adults (Figure 3A), indicating that it might contribute to organism growth. In addition, the expression levels were obviously increased across every stage at 37 °C, while *NlHsp90* transcripts were almost unchanged by cold shock (4 °C) (Figure 3A), which suggested that *NlHsp90* was thermally inducible. In addition, *NlHsp90* was higher expressed in egg and adult stageunder heat shock stress, which was approximately five-fold and three-fold higher than the normal condition. *NlGRP94* expression levels were not significantly different between 27 and 4 °C and were relatively high in adults. When treated with high temperature, a higher expression level was observed in the egg stage (Figure 3A). *NlTRAP1* was abundant in all stages and temperatures (Figure 3A).

For the tissue expression pattern, qPCR results revealed that all three homologs of Hsp90 had specifically high expression in the ovary, followed by the testis (Figure 3B), indicating that they might play important functions in the development of germ cells, especially ovary development and oogenesis.

### 3.4. RNAi and Survival Assay under Normal Conditions

To elucidate the possible influence of three Hsp90 homologs in the development of BPH under normal conditions, second instar and fourth instar nymphs were injected with corresponding dsRNAs and reared at 27 °C. The qRT-PCR results show that the transcript level of each gene was significantly reduced after the RNAi treatments in contrast to the ds*GFP* treatment (Figure 4).

Under normal growth conditions, as shown in Figure 5A,B, when dsRNA was administered during the nymph stage, ds*NlHsp90* and ds*NlGRP94* microinjection led to lethal phenotypes at different levels. Specifically, ds*NlHsp90* treatment caused high mortality, with a 5.6% (2nd treated) and 10.9% (4th treated) survival rate 8 days after injection, compared with an approximate survival rate of 92% for ds*GFP*-treated BPHs. Mortality was slightly lower when the 2nd and 4th instar nymphs were treated with ds*NlGRP94*, approximately 40.9% and 72.4%, respectively. In addition, we found that most of the deaths occurred during nymph-nymph molting or the nymph-adult ecdysis stage, present as the ecdysial line had already split but the insect failed to shed successfully (Figure 6A). However, no distinct phenotypes were observed in the ds*NlTRAP1* treatment.

### 3.5. Effect of BPH Hsp90 Homologs on Oogenesis, Fecundity and Embryogenesis

According to the tissue expression profiles, the expression level of all the three Hsp90 genes was relatively high in the ovary, so we conducted RNAi on the newly emerged females to test the effect on ovary and female fecundity. We dissected the ovaries of virgin females three days, five days and nine days after injection. *NlHsp90* and *NlGRP94* knockdown resulted in oocyte malformation. In the ds*NlTRAP1* and ds*GFP* groups, normally developed oocytes were observed in BPH with a whitish color and slightly banana-like shape (Figure 7C). Specifically, the dysplastic oocytes developed slowly and had malformed, wizened shapes (Figure 7A), and, as a result, nearly all ds*NlHsp90* females failed to produce eggs.

When treated with ds*NlGRP94*, some banana-shaped oocytes could still be observed in the females; however, they were filled with transparent bubbles, and the chorion became rough (Figure 7B,E). In addition, the knockdown of *NlGRP94* severely decreased female fecundity. The oviposition and hatchability experiments showed that only an average of 20 eggs were laid in 5 days, and none of the eggs hatched successfully (Figure 5C,D). In contrast, approximately 95 eggs with 98% hatchability were observed in the ds*GFP*-treated group. Furthermore, the eggs laid by ds*NlGRP94*-treated females failed to form eye pigmentation even in day 7 (Figure 8). The formation of eye pigmentation is a characteristic marker of embryo development that generally appeared on the fifth day after egg deposition. On the contrary, the knockdown of the *NlTRAP1* gene had no effect on female fecundity.

Consistent with the reproductive failure of ds*NlHsp90*-treated females and the decrease in eggs oviposited by ds*NlGRP94*-treated females, the abdomen size was apparently larger in ds*NlGRP94*-treated females than in control females after day 6. In addition, the lateral and intersegmental membranes of the abdomen stretched so much that the cuticle was cracked, leaving some black scars in the abdomen (Figure 6B).

### 3.6. Electron Microscope Observations

To observe the fragile cuticle in ds*NlHsp90-* and ds*NlGRP94*-treated female abdomens, we dissected the dorsal abdominal integument on day six and observed the samples under a transmission electron microscope. After the knockdown of *NlHsp90*, the chitin lamellae of the cuticle were formed normally, but the endocuticle seemed thinner than that of ds*GFP*-treated BPHs (Figure 9B). In ds*NlGRP94*-treated BPHs, chitin lamellae were also formed; however, the lamellae of exocuticles with less sclerotization seemed looser than those of the ds*GFP* group (Figure 9C), suggesting that this gene might affect the structure of the cuticle.

## 4. Discussion

In the present study, we identified three Hsp90 homologs in BPH, cytosolic *Nl*Hsp90, ER-based *Nl*GRP94 and mitochondrial *Nl*TRAP1, among which *Nl*GRP94 and *Nl*TRAP1 were first identified in planthoppers. RNAi-based functional analysis revealed the multifunctional biological role of Hsp90 homologs in BPH.

Sequence analysis showed that all the BPH Hsp90 homologs contain three conserved domains, namely, an N-terminal ATP-binding domain, a middle domain, and a carboxyterminal dimerization domain, as reported in other insects [8]. In addition to the general structure, *NlGRP94* and *NlTRAP1* contain an ER retention signal and mitochondria targeting sequence in the N-terminus, respectively, which further inferred their subcellular location. In contrast to the general KDEL motif at the end of GRP94, the HDEL peptide is the alterative C-terminus in *NlGRP94*. Similar cases were also found in the ER-based Hsp90 of *T. castaneum* and *Drosophila* [16,18]. A study that investigated the comparative genomics and evolution of the Hsp90 family of 32 species across all kingdoms of organisms demonstrated that the KDEL motif is variable across organisms and that the K residue might be replaced by an H, A, E, R, S or N residue [5]. This prompted us to speculate that the HDEL motif might be the more conserved pattern in insects.

*Nl*GRP94 and *Nl*TRAP1 share 46% identity with cytoplasmic *Nl*Hsp90. Phylogenetic analysis showed that the three Hsp90 homologs were clustered into three distinct branches, indicating different functions in response to different stresses in the development of BPH. As *Nl*TRAP1 was well separated from the other two Hsp90s (Class A and C) and clustered with bacterial HtpG (Figure 2), the origin of TRAP1 is still not very clear. A hypothesis has been put forth in a previous study which suggested that cytoplasmic, ER, mitochondrial and chloroplast Hsp90s originated from one ancient eukaryotic ancestor. This view speculated that the ancestor harbored two HtpG genes, one of which evolved into mitochondrial TRAP1 and the other that eventually evolved into the other three forms [28]. Compatible with this view, our results indicate that *Nl*TRAP1 might originate from an HtpG-like ancestor that is distinct from the cytosolic and ER members.

Previous studies demonstrated that cytosolic Hsp90s play pleiotropic roles in insect longevity, spermatogenesis, oogenesis, embryogenesis and postembryonic development [29,30,31,32,33,34]. However, there still exist some details which need further investigation to enhance our comprehension of cytosolic Hsp90s. For example, the expression pattern covering all development stages under environmental stress is still lacking and whether it has other functions in insects. As it is known that cytosolic *NlHsp90* is inducible under heat stress in BPH, in this study, we explored the more specific details both in expression pattern and biological functions. According to the qPCR analysis, we found that cytosolic *NlHsp90* was obviously induced by high temperature at all tested stages (from egg to adult) but could not be induced by cold stress. Interestingly, *NlHsp90* was induced highest in the egg stage and followed by the adult stage under heat shock stress, which may indicate that these two stages were most sensible to the environmental change. Accompanied with a specifically high expression in the ovary, it further prompted us to associate *NlHsp90* with the potential importance in adult development, oogenesis, oviposition and embryonic development. In most cases, nearly all the research results have showed that cytosolic Hsp90 was sensitive to heat. However, there still existed some difference between laboratories on account of diverse experimental conditions, such as temperature gradient, the treating time and the development stages they chose. On one hand, these divergences help to explain the universality of Hsp90 sensitivity, but, on the other hand, it might cause confusion when we make comparison among different studies. Our results might provide more specific information for selecting proper developmental stage in future studies.

In our study, extremely high lethality was observed in nymphs after dsRNA treatment, suggesting that the *Nl*Hsp90 gene was an essential protein in the development of BPH nymphs. Furthermore, the proposed role of Hsp90 in oogenesis during the ovarian maturation of *T. castaneum* was also found in BPH, and the knockdown of *Nl*Hsp90 led to the arrest of ovary development in females and no offspring could be produced. Evidence has shown that a chaperone complex including Hsp90 and Hsc70 is required in vivo for ecdysone receptor activity in *D. melanogaster* [35], which provides a new clue to uncover the underlying mechanism of Hsp90 in contributing to oogenesis. In particular, a new phenotype was observed when the newly emerged females were injected with ds*NlHsp90*. The observation of the thinner abdominal cuticle structure of ds*NlHsp90*-treated females by TEM may imply a novel function of cytosolic Hsp90 in insect cuticles, which serve as the first line of defense against environmental factors, such as temperature, humidity, oxygen concentration, ultraviolet, heavy metal and pathogens etc. Our new discovery may offer clues to explore the involvement of cytosolic Hsp90 in insect cuticle structure and function under environmental stress.

GRP94 serves an essential function during a specific, early developmental stage in different metazoan species. For example, the loss of function of GRP94 in *C. elegans* caused development arrest at an early larval stage and prevented development into mature adults [15]. In *Drosophila*, Gp93 (GRP94 homolog) is not only indispensable for larval development but is also required for midgut epithelial homeostasis [16]. However, there are still limited studies evaluating the function of the GRP94 ortholog in insects, and the functions of GPR94 in Hemiptera have seldom been explored. In the present study, the RNAi-mediated knockdown of *NlGRP94* in early- and late-stage BPH nymphs led to lethality, but lethality was lower when late-stage insects were injected. This observation suggested that the basic function of GRP94 in the early developmental stage might be evolutionarily conserved.

Unlike cytosolic Hsp90, GRP94 is usually thought to be induced by physiological ER stress rather than high temperatures or other stresses that are unique to the cytosol [36]. However, our qPCR results show significantly increased *NlGRP94* transcripts in the egg stage after heat shock at 37 °C. Such high temperature stress-induced expression was also found in grass carp (*Ctenopharyngodon idella*) and *T. castaneum* [18,37]. Nevertheless, *NlGRP94* gene expression was not induced either by heat or cold treatment in other developmental stages except for eggs, which indicated a special role that *NlGRP94* might play in this most fragile BPH stage. Under normal conditions, a relatively high expression level of *NlGRP94* was observed in adults and specifically expressed in the ovary. By using RNAi, novel phenotypes were discovered in BPH oogenesis and female fecundity. The defective ovary and decreased BPH fecundity observed after *NlGRP94* knockdown were in accordance with qPCR results. In detail, when the newly emerged females were injected with ds*NlGRP94*, the ovariole was abnormal with a rough egg in the base cell and filled with transparent bubbles. GRP78, another major ER stress protein that is thought to work in concert with GRP94 [38], has been reported to be involved in insect fat body cell homeostasis and vitellogenesis and a regulatory target of juvenile hormone [39]. Therefore, one possible explanation is that the decreased expression of GRP94 may affect its relationship with GRP78, which finally influences fat body cell homeostasis and vitellogenesis.

Studies in mouse and *T. castaneum* demonstrated that ER-based Hsp90 was required during early embryo development [18,40]. In our study, there was no visible difference in the early development of eggs until the control group formed eye pigmentation on the fourth day. These results suggest that *NlGRP94* may regulate embryogenesis and may be related to the later rather than early period of embryonic development in this species. 

What interested us most was that the TEM results show a disordered abdominal integument structure, which might account for the scar observed in the cuticle after *NlGRP94* knockdown. This possible function for the cuticle stability of *NlGRP94* was first discovered. Together with the thinner abdominal cuticle structure observed in ds*NlHsp90*-treated females, we can conclude that Hsp90s might have a special function in the cuticle structure and formation. And the relationship between Hsp90 and GRP94 needs further exploration.

Research on insect TRAP1 is very limited. In our study, we first analyzed the expression pattern of *NlTRAP1* across different developmental stages and tissues. The qPCR results show that *NlTRAP1* was expressed across all developmental stages and could not be induced by either heat or cold stress. Similar with *NlHsp90* and *NlGRP94*, *NlTRAP1* was highly expressed in the ovary. However, no obvious phenotype was observed after RNAi at different BPH stages, especially in females, indicating that the loss of function of *NlTRAP1* under normal growth conditions had no significant effect. As a mitochondrial protein, it was reported to protect cells from oxidative stress and apoptosis. We wonder whether *NlTRAP1* can function under oxidative stress. So, further research could focus on the possible function of TRAP1 under oxidative stress.

Extraordinary experimental efforts have been made in illustrating the role of Hsp90s in physiological processes such as protein folding, cell signaling and apoptosis, and human diseases, such as cancer and neurodegenerative diseases [2]. However, the type of diseases they participate and the underlying mechanisms are still lacking. Consequently, our observations of high expression level in ovary of all the three Hsp90s and essential role in oogenesis and embryogenesis, we hope, might provide clues to explore the function of human Hsp90 isoforms in ovary and embryonic development under normal physiology and thermal or oxidative stress.

## 5. Conclusions

In summary, we identified and characterized all three Hsp90 homologs in BPH, *Nl*Hsp90, *Nl*GRP94 and *Nl*TRAP1. Sequence analysis not only revealed the general structure of three conserved domains but also the distinct characteristics of different members located in the cytosol, ER and mitochondria. Cytosolic *NlHsp90* was significantly induced by a high temperature throughout development, and ER-based *NlGRP94* was inducible only in the egg period at 37 °C, while *NlTRAP1* was not inducible by temperature stress. The RNAi-mediated knockdown of *NlHsp90* revealed the essential role of this gene in nymph development and oogenesis under physiological conditions. *NlGRP94* was also required during the early developmental stage and played an important role in oogenesis, female fecundity and late embryogenesis. Notably, *NlHsp90* and *NlGRP94* were likely involved in the cuticle structure of female BPH. Consequently, these findings reveal the ubiquitous expression, sequence conservation and multifunctional functions of Hsp90 homologs in the brown planthopper and thus provide clues for further research into Hsp90 in human beings/other animals.

## Figures and Tables

**Figure 1 genes-11-01074-f001:**
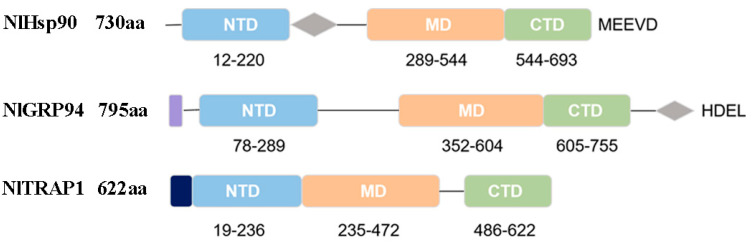
Schematic diagram of the domains of the three brown planthopper (BPH) Hsp90 homologs. *Purple*, endoplasmic reticulum (ER) retention sequence; *black*, mitochondria targeting sequence; *blue*, N-terminal domain (NTD); *gray*, charged linker region; *orange*, middle domain (MD); *green*, C-terminal domain (CTD).

**Figure 2 genes-11-01074-f002:**
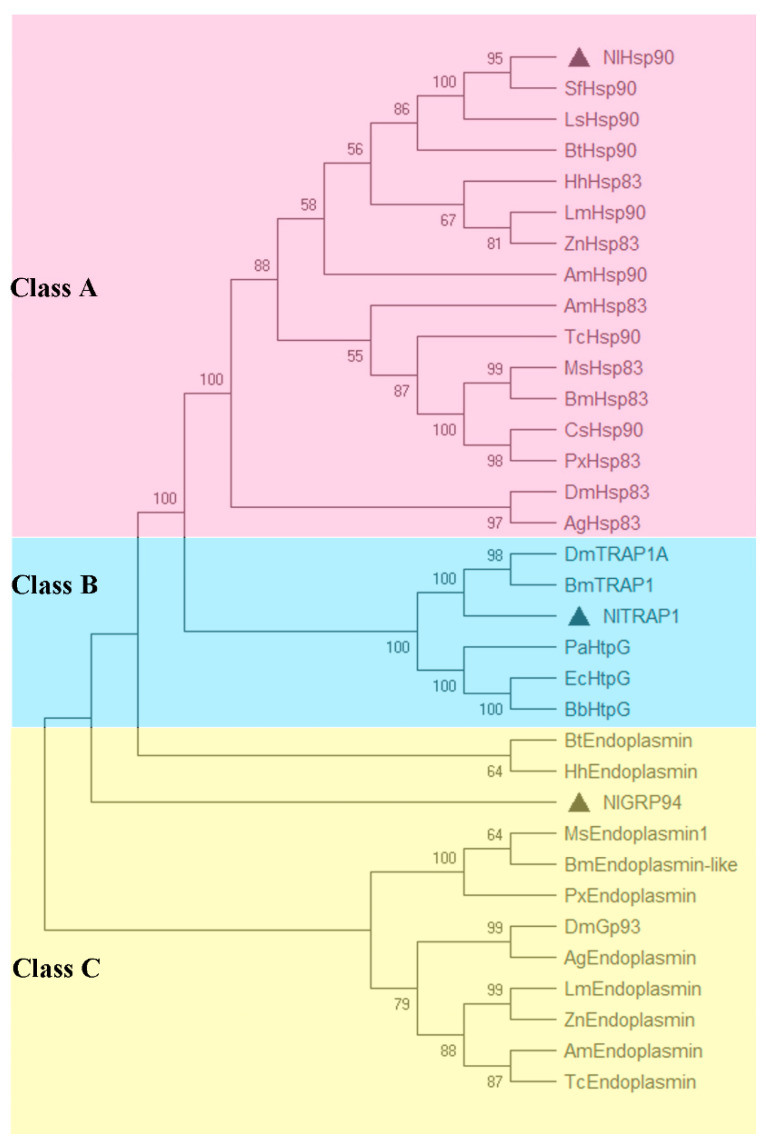
Phylogenetic analysis of the Hsp90 family members in BPH. The phylogenetic tree was constructed based on 34 Hsp90s from 18 species (see Appendix A) using the neighbor-joining method, and bootstraps were set with 1000 replications. The Hsp90 members of BPH are shown with black triangles.

**Figure 3 genes-11-01074-f003:**
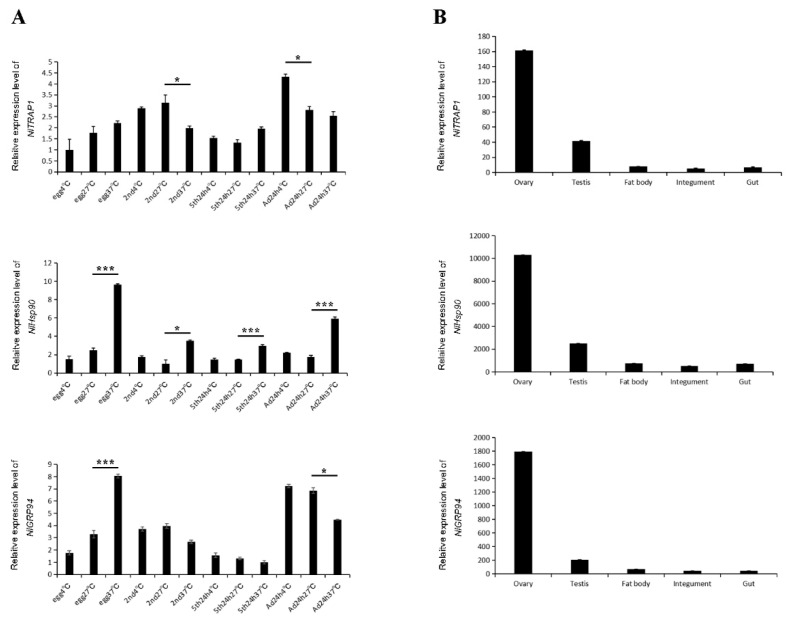
Developmental and tissue-specific expression patterns of *NlHsp90*, *NlTRP94* and *NlTRAP1*. (**A**) Developmental profile of three genes under different temperatures. Total RNA was isolated from eggs, second instar nymphs, fifth instar nymphs and adults 24 h after molting at 4 °C, 27 °C and 37 °C for 1 h. (**B**) Tissue-specific expression of three genes. Total RNA was isolated from ovary, testis, fat body, integument and gut of BPH. Real-time qPCR (RT-qPCR) combined with the 2^−ΔΔCt^ method was used to evaluate the relative expression levels of target genes. The results are presented as the mean ± SEM (standard error of mean). * *p* < 0.01, *** *p* < 0.001 (Student’s *t*-test). Bars represent the SEM derived from three independent biological replicates.

**Figure 4 genes-11-01074-f004:**
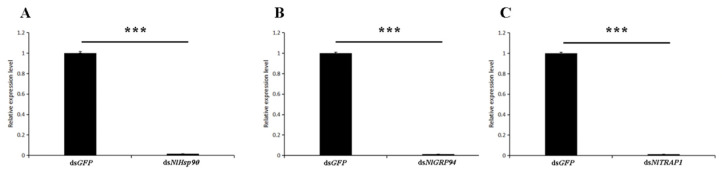
Transcript levels of *N. lugens* Hsp90s ((**A**). *NlHsp90*, (**B**). *NlTRP94* and (**C**). *NlTRAP1*) after double-stranded RNA injection. Three days after injection, 6 to 8 insects were collected randomly to evaluate the RNAi efficiency. RT-qPCR and the 2^−ΔΔCt^ method were used to measure the relative transcript levels. The results are presented as the mean ± SEM. *** *p* < 0.001 (Student’s *t*-test). Bars represent the SEM derived from three independent biological replicates.

**Figure 5 genes-11-01074-f005:**
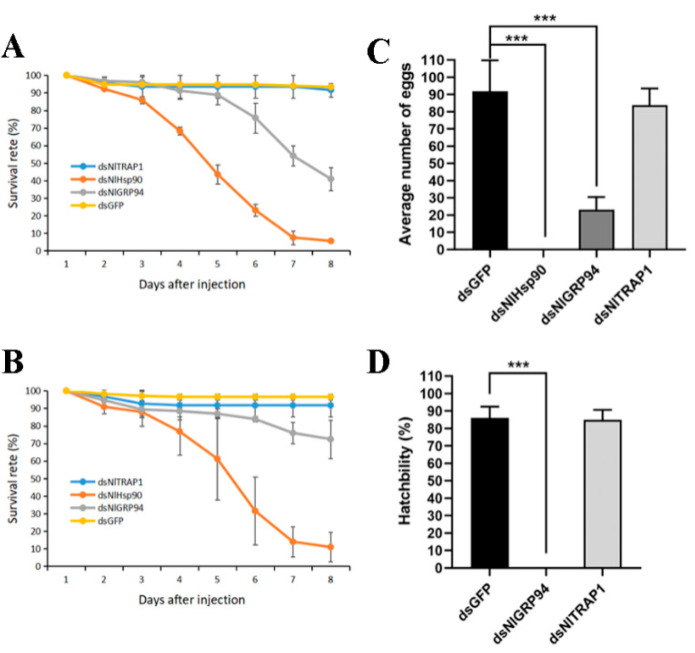
Effects of dsRNA treatments on nymph survival and female fecundity. (**A**) The survival rate of BPH 8 days after RNAi when 2nd instar nymphs were injected with dsRNA. ds*GFP* was used as the negative control in each group, *n* = 100. Three biological replicates were performed (data are shown as means ± SEM). (**B**) The survival rate of BPH eight days after RNAi when 4th instar nymphs were injected with dsRNA. (**C**) Average number of eggs laid by each female after different RNAi treatments. (**D**) Egg hatchability after RNAi treatment. Ten repetitions were performed. The results are shown as the mean ± SEM. *** *p* < 0.001 (Student’s *t*-test).

**Figure 6 genes-11-01074-f006:**
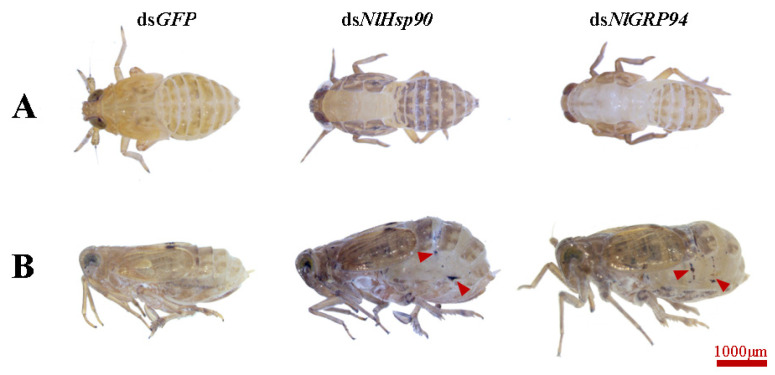
Phenotypes of *N. lugens* injected with ds*NlHsp90* and ds*NlGRP94*. (**A**) Influence of ds*NlHsp90* and ds*NlGRP94* on the development of nymphs. Most nymphs died during molting. (**B**) The phenotype of ds*NlHsp90* and ds*NlGRP94* females 6 days after injection, with stretched intersegmental membranes and scars in the abdomen. The black scar in the abdomen is indicated by arrowheads. ds*GFP* was injected as a negative control.

**Figure 7 genes-11-01074-f007:**
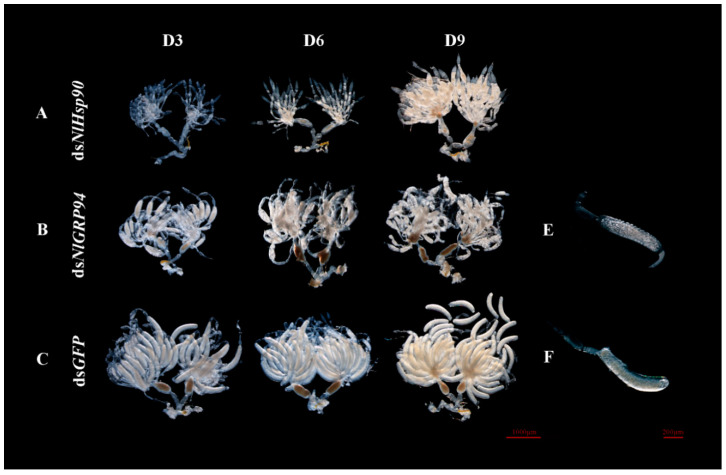
Phenotypes of ovary development upon gene knockdown. (**A**–**C**) Ovaries dissected from ds*Hsp90-*, ds*NlGRP94-* and ds*GFP*-treated females at 3, 6 and 9 days after adult eclosion. (**D3**), (**D6**) and (**D9**) represent day three, day six and day nine of the ovary. (**E**) An oocyte from the ovaries of ds*NlGRP94*-treated females. (**F**) An oocyte from the ovaries of ds*GFP*-treated females.

**Figure 8 genes-11-01074-f008:**
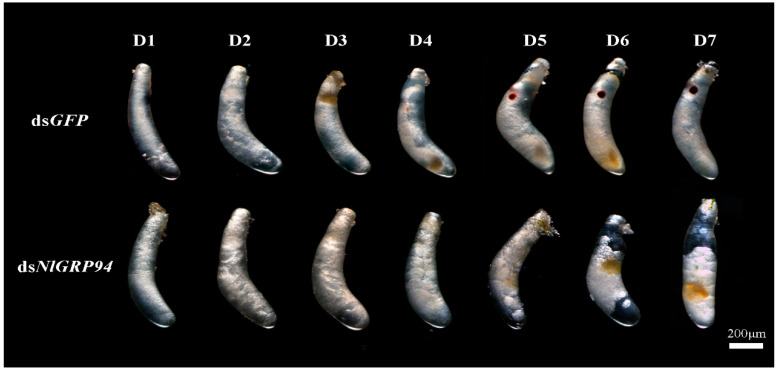
Embryonic development after ds*NlGRP94* injection. The newly emerged females were treated with dsRNA and were mated with males. After oviposition, the eggs were dissected from the rice stem and observed every day. The development of eggs laid by ds*GFP*-treated females used as the parallel control.

**Figure 9 genes-11-01074-f009:**
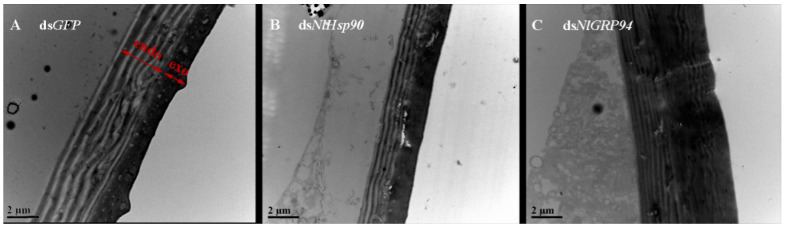
Transmission electron microscopy observation of the dorsal abdominal integument of the sixth-day females in (**A**) the ds*GFP*-injected group, (**B**) the ds*NlHsp90*-injected group and (**C**) the ds*NlGRP94*-injected group.

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
