# Peer review of "Identification and Characterization of Three Heat Shock Protein 90 (Hsp90) Homologs in the Brown Planthopper"

_genes, 2020, doi:10.3390/genes11091074_

Round 1

Reviewer 1 Report

"Identification and Characterization of 3 Heat Shock Protein 90 (HSP90) Homologs in the Brown 3 Planthopper" is an interesting research article that investigates the expression of different isoforms of HSP90 and their biological role in insects. The introduction is well written and allows a quick understanding of the specific field of research. The overall level of English is adequate, the study design is intriguing and the bibliography is consistent with the topic.

1) A "Statistical analysis" paragraph should be added to the materials and methods section.

2) Statistical analysis and significance should be added in Figure 3 and 4 graphs and description.

3) The results in figure 3 A underlines HSP90 sensitivity to heat. Authors should consider to add in the discussion how HSP90 sensitivity can result in different results between laboratories with diverse experimental conditions and subsequent limitations.

4) Authors should consider to translate, when possible, the meaning of this results to the human cytoplasmic and mitochondrial isoforms and their role in physiology and diseases.

Author Response

Comment: 1) A "Statistical analysis" paragraph should be added to the materials and methods section.
Response: Thanks. We have added a "Statistical analysis" paragraph in the materials and methods section in L153-156 as follows:
2.6. Statistical analysis
Data are presented as the mean ± SEM. Statistical analysis was performed using Microsoft EXCEL and GraphPad Prism 8.3.0. Two-tailed Student’s t-test was used to compare the difference between two groups. The significance level was set at *p < 0.05 and ***p < 0.001.

Comment: 2) Statistical analysis and significance should be added in Figure 3 and 4 graphs and description.
Response: Thanks. We have added statistical analysis and significance in Figure 3 (between each normal temperature and thermal/cold stress) and 4 graphs (between each treatment and the control) and related descriptions.
Figure 3:

“The results are presented as the mean ± SEM. *P < 0.01, ***P < 0.001 (Student’s t-test).” was added in the description.

Figure 4:

“The results are presented as the mean ± SEM. ***P < 0.001 (Student’s t-test).” was added in the description.

Comment: 3) The results in figure 3 A underlines HSP90 sensitivity to heat. Authors should consider to add in the discussion how HSP90 sensitivity can result in different results between laboratories with diverse experimental conditions and subsequent limitations.
Response: In L 345-353, we added the discussion as follows: In most cases, nearly all the research results have showed that cytosolic Hsp90 was sensitive to heat. However, there still existed some difference between laboratories on account of diverse experimental conditions, such as temperature gradient, the treating time and the development stages they chose. On one hand, these divergences help to explain the universality of Hsp90 sensitivity, but on the other hand, it might cause confusion when we make comparison among different studies. Our results might provide more specific information for selecting proper developmental stage in future studies.

Comment: 4) Authors should consider to translate, when possible, the meaning of this results to the human cytoplasmic and mitochondrial isoforms and their role in physiology and diseases.
Response: In L416-422, we added the discussion as follows: Extraordinary experimental efforts have been made in illustrating the role of Hsp90s in physiological processes like protein folding, cell signaling and apoptosis, and human diseases, such as cancer and neurodegenerative diseases [2]. However, the type of diseases they participate and the underlying mechanisms are still lacking. Consequently, our observations of high expression level in ovary of all the three Hsp90s and essential role in oogenesis and embryogenesis, we hope, might provide clues to explore the function of human Hsp90 isoforms in ovary and embryonic development under normal physiology and thermal or oxidative stress.

In Line 435-437, we added “...multifunctional functions of Hsp90 homologs in the brown planthopper and thus provide clues for further research of Hsp90 in human being/other animals.”

Reviewer 2 Report

The paper by Chen et al. describes the identification and characterization of Hsp90 proteins encoded in the genome of N. lugens. The authors used sequences of Drosophila Hsp90 genes to identify and isolate Hsp90 genes from the N.l. genome. By phylogenetic analysis they show the genes are homologous to and can be classified with the Eukaryotic cytosolic, ER and mitochondrial families. They use a combination of monitoring expression patterns, disrupting of expression, dissecting organs and observing structures by electron microscopy to characterize importance and function of the three Hsp90s. The data imply that cytosolic Hsp90 is essential for viability and that cytosolic and ER Hsp90s are crucial for ovary and oocyte development. Their EM analysis indicates Hsp90 function is important for development of abdomen cuticle. The work is well-controlled, the data are convincing, and the interpretations and conclusions are reasonable and supported by the data. Overall the study provides useful information for vital proteins of an agriculturally important insect that offer new insight into functions Hsp90 and point to possible targets for controlling the organism. 

Author Response

We appreciate the reviewer for his/her positive comments.